# Analysis of Missingness Scenarios for Observational Health Data

**DOI:** 10.3390/jpm14050514

**Published:** 2024-05-11

**Authors:** Alireza Zamanian, Henrik von Kleist, Octavia-Andreea Ciora, Marta Piperno, Gino Lancho, Narges Ahmidi

**Affiliations:** 1Department of Computer Science, TUM School of Computation, Information and Technology, Technical University of Munich, 85748 Munich, Germany; henrik.vonkleist@helmholtz-munich.de; 2Fraunhofer Institute for Cognitive Systems IKS, 80686 Munich, Germany; octavia.ciora@iks.fraunhofer.de (O.-A.C.); marta.piperno@tum.de (M.P.); gino.lancho@iks.fraunhofer.de (G.L.); narges.ahmidi@gmail.com (N.A.); 3Institute of Computational Biology, Helmholtz Center Munich, 80939 Munich, Germany

**Keywords:** missing data analysis, observational health data, missingness scenarios, missing data assumptions, missingness distribution shift

## Abstract

**Simple Summary:**

This paper argues the importance of considering domain knowledge when dealing with missing data in healthcare. We identify fundamental missingness scenarios in healthcare facilities and show how they impact the missing data analysis methods.

**Abstract:**

Despite the extensive literature on missing data theory and cautionary articles emphasizing the importance of realistic analysis for healthcare data, a critical gap persists in incorporating domain knowledge into the missing data methods. In this paper, we argue that the remedy is to identify the key scenarios that lead to data missingness and investigate their theoretical implications. Based on this proposal, we first introduce an analysis framework where we investigate how different observation agents, such as physicians, influence the data availability and then scrutinize each scenario with respect to the steps in the missing data analysis. We apply this framework to the case study of observational data in healthcare facilities. We identify ten fundamental missingness scenarios and show how they influence the identification step for missing data graphical models, inverse probability weighting estimation, and exponential tilting sensitivity analysis. To emphasize how domain-informed analysis can improve method reliability, we conduct simulation studies under the influence of various missingness scenarios. We compare the results of three common methods in medical data analysis: complete-case analysis, Missforest imputation, and inverse probability weighting estimation. The experiments are conducted for two objectives: variable mean estimation and classification accuracy. We advocate for our analysis approach as a reference for the observational health data analysis. Beyond that, we also posit that the proposed analysis framework is applicable to other medical domains.

## 1. Introduction

Healthcare data encompass a wide range of variables related to various diseases and health conditions collected from different facilities under the supervision of distinct and even contrasting guidelines. It is, therefore, naive to expect a medical dataset in which a sufficient number of informative variables are available for all patients. This is why data-driven research in healthcare almost always faces the challenges of missing data.

As two out of many existing approaches, data scientists may choose to simply discard the incomplete data samples and use only the complete ones for analysis (complete-case analysis) or utilize complex deep learning models to fill in the missing entries and create a complete semi-synthetic dataset (imputation). In any case, the reliability of the methods depends on the nature of the missingness problem, e.g., how a variable distribution changes when the variable is observed or missed, or why physicians decide to measure a variable for a patient and not the other. Inevitably, we must make assumptions about these questions, which are often not testable using the data alone [1,2]. Hence, we can trust the analysis result only if the assumptions are made explicit, the method conforms to them, and the sensitivity of the results to departures are investigated [1].

Even though the scholarly discourse on missing data consistently highlights this concern [3,4,5,6,7], surprisingly, the applied machine learning (ML) research in healthcare often suffers from ambiguous missingness problem statements [6] and the lack of transparent reports of assumptions for methods which are detached from the reality of healthcare facilities [4]. The complete randomness of missingness is assumed even though observations in healthcare facilities are clearly conducted by guidelines and protocols [6,7]. Missing data methods are often chosen inconsistently when training and evaluating prediction models [8]. The problem intensifies, as the survey and review articles on missing data in healthcare [5,9,10,11,12] are often limited to merely imputing the medical datasets without the scrutiny of assumptions.

We believe that the remedy is a formalization of the events and scenarios within healthcare facilities that result in missing data. Having a curated catalog of common and influential scenarios along with their theoretical implications for the missing data analysis, medical data scientists can examine the data collection environment of their datasets. They can determine which scenarios apply to their situation, and tailor their analysis accordingly.

In this regard, this paper introduces a framework for identifying missingness scenarios in a data collection environment and translating them to the language of missing data theory. We apply this framework to the healthcare domain, introducing and analyzing ten fundamental missingness scenarios for observational data in healthcare facilities.

### 1.1. Contribution and Scope

Addressing a theory–application gap in healthcare, this paper falls under the translational research category. We mainly focus on missingness scenarios in healthcare facilities such as clinics and hospitals. We exclude planned clinical studies, as they comprise considerably different scenarios (e.g., case drop-out and planned missingness). We achieve the goal of the paper through the following steps:We introduce an analysis framework for identifying and analyzing the missingness scenarios in different domains of application.We introduce ten fundamental and prevalent scenarios in healthcare facilities that lead to the observation, recording, or missingness of data. Table 1 gives an overview of the scenarios.To make the theory–application connection, we introduce theoretical inquiries to be made about each missingness scenario. Table 2 gives an overview of the inquiries.For each scenario, we make the inquiries above and analyze the theoretical implications, along with various examples from the medical data analysis literature.To demonstrate the effect of domain-informed assumption on the method reliability, we perform a simulation study, showing how domain-agnostic analysis may lead to different levels of bias depending on the active scenarios and different estimands.

We propose this paper as a reference point for correct data analysis and reporting using observational health data. The methodology of this paper is not limited to the selected missingness scenarios and can be applied to less common yet equally essential scenarios that the readers may encounter. For the scope of the paper, we mainly focus on problem formulation and missing data assumptions, minimally going into detailed discussions about the implementation of missing data algorithms and methods. Nevertheless, when required, we provide sufficient references to influential works in the missing data literature.

### 1.2. Structure

To present a clear image of the problem at hand and the existing research gap, we introduce the general ideal approach to missing data analysis and highlight the influential body of works regarding this approach in Section 2. The methodology of the paper, presented in Section 3, is divided into three separate parts: After introducing the analysis framework in Section 3.1, first, we identify the key missingness scenarios in healthcare facilities in Section 3.2 and then analyze the scenarios within the framework of missing data theory in Section 3.3. We conduct experiments to show the significance of domain-informed missing data analysis in Section 4. Discussions, limitations, and proposals for future works are presented in Section 5.

## 2. Theoretical Background

### 2.1. An Ideal Approach to Missing Data Analysis

The ideal approach to the analysis with missing data comprises a correct formulation of the analysis target, a correct estimation of it, and finally investigating the effect of possible errors in the formulation and estimation steps. This approach, established and adopted in the missing data literature [2,13] makes a common skeleton for the missing data methodologies. It is only in the details of each step of this approach where the methodologies differ. We elaborate on the approach, depicted in Figure 1, as follows:Analysis begins by formulating the objective (e.g., risk factors for a disease) as a mathematical expression, namely an estimand, which should be learned using incomplete data.Next, assumptions about the missingness status of the variables in the estimand are collected. Such information could include, for example, how other variables influence the missingness in another variable, or which model is the most suitable for describing the chance of observation or missingness. As depicted in Figure 1, the assumptions must be directly informed by the actual setting of the problem at hand.In light of the assumptions, it is determined whether and how the estimand can be learned from the incomplete data. This step is referred to as *identification* (or *recovery*), and the estimand is called *identifiable* if the answer to the identification step is positive. The identification property is the direct product of the objective and the problem setting. It may be the case that an objective cannot be recovered from the available data. A trivial example of unidentifiability is to estimate the mean value of a variable while it is completely unavailable.The next step is to learn the estimand using the available data, e.g., by employing ML methods. This step is referred to as *estimation*. The right method and learning setting is also influenced by the problem assumptions.Finally, to increase the reliability and robustness of the results, the results’ sensitivity to model perturbations and violation of assumptions is measured (sensitivity analysis). Similar to steps 3 and 4, this step is also influenced by the assumptions when choosing meaningful perturbation ranges according to the problem at hand.

### 2.2. Research Gap

Related to the introduced ideal approach in the previous section, Table 3 presents examples of related works supporting each analysis step: (1) correct formulation of different estimands, (3) identification theory for missing data problems, (4) efficient estimation for missing data, and (5) sensitivity analysis.

Regarding step 2 (assumption specification), which is the focus of this paper, cautionary articles exist for advocating domain-informed analysis [3,5,7], and providing high-level strategies for correct interpretation and reporting of missingness scenarios [4,14]. Several papers have performed such domain-informed analyses (for example, Mirkes et al. [15] and Millard et al. [16]), yet only for specific diagnoses (trauma and COVID-19), limited to unique missingness complications (selection bias) and estimands (outcome prediction). To our knowledge, no comprehensive taxonomy and analysis of missingness scenarios exists for observational data in healthcare facilities.

Our paper is particularly inspired by Moreno-Betancur et al. [17] and Marino et al. [18]. These works develop general guidelines for treating missing data in epidemiology and clinical (point-exposure) studies, mainly focusing on missingness scenarios and circumstances that suit these settings. To name a few, they consider the case dropout, where participants are removed from the remaining of the study if they miss a measurement session, and the assumption of no hidden confounders, which is the product of a curated study design. As outlined in Section 3.2, different and more diverse reasons for missingness exist in healthcare facilities, where patient admission, data observation, and data collection, do not always follow a designed plan as they would in clinical studies.

Despite this difference, their motivation and approach for identifying and analyzing missingness scenarios remain relevant for observational health data—a foundational principle upon which this paper is built. The increasing need for developing missing data methods is reflected in the findings of the survey and systematic review papers, including that of Ismail et al. [12], which reported a three-fold increase in the publication of imputation techniques for machine learning algorithms alone. This shows the need for a clear guideline for reliable missing data analysis in all healthcare domains, including that of observational data within healthcare facilities.

**Table 3 jpm-14-00514-t003:** Examples of literature supporting the steps of the missing data analysis process, presented in Figure 1.

Step	Subject Matter	References
1	Estimand formulation under missing data	[2,19,20,21,22]
2	Domain-informed missing data formulation and assumptions *	[15,16]
3	Missing data identification theory	[2,23,24,25,26,27,28]
4	Estimation with missing data	[20,21,29,30,31,32]
5	Sensitivity analysis for unidentifiable missingness	[33,34,35,36]

* Research gap.

## 3. Methods

### 3.1. Analysis Framework

Developing the missingness scenario-to-theory mapping would first require knowing what lies at both ends of the mapping. Then, the mapping itself must be developed, determining the theoretical implications of every scenario.

To identify scenarios, first, we investigate the steps through which the data end up in the analysis dataset:First, a piece of evidence is obtained through observation or measurement;After observation, the data must be recorded in the database which is the source of the analysis dataset;Finally, the recorded data must be retrieved for creating the dataset and kept (not removed) for data analysis.

An interruption in any of these steps will lead to missing data (unavailability), i.e., when data are not observed, not recorded, or not included for the analysis.

Depending on the process, these three steps may occur naturally until a specific event interrupts them. For instance, occupation information is normally obtained and recorded unless patients refuse to provide it, while MRI images are generally not available for patients unless the attending physicians order them. Therefore, depending on the features of the data and the data collection environment, we must investigate either the scenarios that lead to the interruption of usual data availability, or actively making the data available despite the default conditions.

Another point of investigation concerns the *agents* that are responsible and directly influence data observation, recording, and inclusion. We identify four agents for the case study of healthcare facilities: (1) patients, (2) attending physicians and medical staff, (3) healthcare facility elements, including the medical devices, medical data software, etc., and (4) medical data scientists, analyzing the resulting datasets.

Cross-investigation of the data availability steps results in identifying the missingness scenarios. In particular, one shall study whether and how an agent influences data observation, recording, and inclusion in the dataset. This includes either the interruption of a usual availability process, or actively deciding to observe or collect the data. The cross-investigation can be performed for any type of case study. In the scope of this paper, we apply it to the domain of observational data from healthcare facilities. Section 3.2 provides the result of the investigation. As a summary, Table 1 presents the identified scenarios.

Once the scenarios are extracted, we must scrutinize each to find out their implication for many assumptions that restrict the effectiveness of the analysis steps, namely, identification, estimation, and sensitivity analysis. We do so by collecting a set of inquiries made about each scenario, which influence the assumptions in the analysis step. Clearly, different methods may require different inquiries. Therefore, the analysis presented in this paper might slightly differ if the reader considers different identification, estimation, or sensitivity analysis methods. Within the scope of this paper, we choose widely utilized and well-established methodologies for the sake of wider applicability; the methodologies are identification theory for graphical models, inverse probability weighting estimation, and exponential tilting sensitivity analysis. They will be introduced in detail in Section 3.3. As a summary, Table 2 presents the inquiries analyzed in this paper. Figure 2 presents a schematic of the analysis framework.

### 3.2. Missingness Scenarios in Healthcare Data

Observational data from healthcare facilities, such as clinics and hospitals, comprise information about outpatient and inpatient (hospitalization) visits. Variables in healthcare data include patient demographic information (e.g., age and gender), medical history, signs and symptoms (e.g., blood pressure value and pain symptom), lab tests (e.g., blood chemistry test), diagnoses, and prescribed medications. The variable list is extended for inpatient visits, including higher resolution observations and prescription information (e.g., oxygen saturation from bedside monitoring, and the input/output chart). In addition, new variables are collected during different hospitalization modes, such as ICU hospitalization. For more information about collected observations in healthcare facilities, see the documentation of publicly available datasets such as the MIMIC-IV electronic health record dataset [37].

Health variables are available through various data observation, recording, and inclusion scenarios that are influenced by the four agents introduced above. In this section, we explore ten fundamental and prevalent scenarios that drive the data-availability steps in healthcare facilities. For each scenario, we provide real-world examples within the text, as well as in Appendix A. The majority of examples are extracted from the clinical prediction model (CPM) literature, introduced by Tsvetanova et al. [8]. CPMs make a crucial portion of AI applications in healthcare, and this work has performed an extensive investigation to highlight highly relevant and well-known CPM models.

#### 3.2.1. Scenarios Related to Patients

By default, data are only collected during a patient’s visit to a healthcare facility, resulting in a gap in data for the time between visits. At the same time, the sub-population that has not visited the healthcare facility is not observed at all. These are situations where we inevitably encounter the missing data problem unless additional data sources are used to complement the primary dataset.

It is naive to assume that visits are decided randomly and unrelated to the patient’s health condition. This motivates the following key missingness scenario:

**Scenario 1** (Patient complete non-visit)**:** Sub-populations with no healthcare facility visits during data collection are not included in the dataset.

In Scenario 1, a specific sub-population is completely missing, e.g., due to health status. For instance, healthy people with no serious health complications infrequently visit clinics with different intentions such as preventive check-ups. Likewise, the data of patients deceased before any visit (e.g., dead-on-arrival) are often absent from the facility database. Other factors, such as socioeconomic status, can also influence the non-visit. References to Scenario 1 are presented in Section A.1.

Scenario 2 describes another type of non-visit, namely missing follow-ups for patients with at least one recorded visit.

**Scenario 2** (Missing follow-up visit due to health status)**:** Patients may miss a follow-up visit due to death, facility transfer, or if they decide not to continue the treatment.

The difference between Scenarios 1 and 2 lies mainly in the reasons for missingness; patients have potentially different reasons not to visit a healthcare facility for the first time, or to drop the follow-up visits, possibly despite the physician’s recommendations. References to Scenario 2 are presented in Section A.2.

Health status factors influence not only the visits but also measurements during the visits. As highlighted by Scenario 3, location transfer within the facility due to health conditions influences the observed variables.

**Scenario 3** (Missing measurements due to health-related events during hospitalization)**:** Observations may be interrupted or limited by extreme health conditions or transfer to a different location.

In Scenario 3, observations may be interrupted due to events such as the occurrence of *code blue* and the resulting disconnection of devices for resuscitation [11,38], or patient transfer, e.g., to the operation room or ICU ward [14]. These events may also lead to observing new health variables that had not been recorded prior to the event, e.g., monitoring during operation [37]. References to Scenario 3 are presented in Section A.3.

Another reason for missingness in a variable is the patient’s refusal to take a test or consent to data sharing as stated by Scenario 4.

**Scenario 4** (Patient’s refusal)**:** Patients may actively refuse specific observations or decline consent to data sharing.

Overall, patients’ personal decisions, whether for medical (e.g., pain intolerance) or non-medical (e.g., fear of examination) reasons, may induce missingness. References to Scenario 4 are presented in Section A.4.

#### 3.2.2. Scenarios Related to Physicians

During a visit, the attending physicians decide which variables to observe. Bickley and Szilagyi [39] describe the examination and diagnosis practice as a step-by-step process in which physicians use basic observations such as vital signs and symptoms to form a first diagnostic belief, referred to as *impression*. To prove or rule out the possible diagnoses, physicians then order more specific, expensive, and sometimes invasive tests. Therefore, as Scenario 5 states, the primary reason for taking or skipping a measurement is the diagnostic information it provides, compared to the cost of observation (e.g., monetary, time, and/or harm to the patient.).

**Scenario 5** (Missing measurements due to diagnostic irrelevance)**:** Variables that are less relevant to the physician’s impressions are less likely to be observed.

Scenario 5 concerns diagnostic flowcharts and score systems in a dataset (see Elovic and Pourmand [40]). These provide rules for selecting the following observation until the final diagnosis. Nevertheless, observation patterns may not entirely reflect one particular guideline, as many guidelines are used during the data collection phase within a cohort, and other scenarios also affect the data.

One should note the implications and subtle differences between these tools when conducting a missingness analysis. For example, in flowcharts, the value range for the parent variable(s) determines the next observation. In contrast, in a score system, the cumulative score of all related variables determines whether more observations are necessary for concluding the decision [41]. References to Scenario 5 are presented in Section A.5.

#### 3.2.3. Scenarios Related to Healthcare Facilities

Measurement decisions are not only determined by physicians but also by protocols and guidelines in healthcare facilities as stated by Scenario 6.

**Scenario 6** (Missing measurements outside protocol requirements)**:** Data collection protocols decide the measurements in different conditions during hospitalization.

For instance, hospital protocols may mandate specific data (e.g., demographic information and basic blood tests) to be collected upon admission. Similarly, there are measurements only performed in particular conditions, e.g., pre- and post-surgical measurements. It is, therefore, crucial to consider the role of protocols within healthcare facilities when investigating the causes of missingness or observation of variables. References to Scenario 6 are presented in Section A.6.

Scenarios 5 and 6 assume that measurements can always be taken if required. While this may generally be true, especially for routine tests, a measurement may sometimes be hindered by the unavailability or shortage of necessary resources. Diagnostic tests may be dropped or delayed for a patient due to prioritization in long waiting queues or temporary unavailability of equipment or staff. Scenario 7 describes the situation where measurement orders were not realized despite physicians’ decisions.

**Scenario 7** (Unavailability or shortage of resources)**:** The physician’s order for observation may not be realized due to unavailability or shortage of resources.

References to Scenario 7 are presented in Section A.7.

Another assumption for Scenarios 5 and 6 which does not always hold is that the measurements and physicians’ observations are all recorded in the database. As stated by Scenario 8, variables might be observed and influence medical decisions, yet they are withheld from the dataset.

**Scenario 8** (Unrecorded observations)**:** Some variables are not recorded in the database or used in the data analysis, even though they have been observed and relied upon in medical practice.

There might be aspects characterizing the overall patient’s health, which are not explicitly recorded but implicitly considered in the decision-making process. In addition, certain modalities of data may not be efficiently recorded or integrated into the medical record. Further, some modalities, such as textual data, may be excluded from data analysis due to complexity. All these reasons are mainly determined by the quality of data collection software in healthcare facilities, the physician’s style of practice, and the choice of data modalities for analysis. References to Scenario 8 are presented in Section A.8.

#### 3.2.4. Scenarios Related to Data Pre-Processing

For the analysis of collected and recorded data, the first step is dataset selection and pre-processing. Depending on the analysis objective, data scientists apply inclusion/exclusion criteria based on demographic information, patient cohort, or variable availability. As stated by Scenario 9, this should be considered a missingness scenario induced in the data analysis step.

**Scenario 9** (Data sample omission based on inclusion/exclusion criteria)**:** Samples are included or excluded depending on data and missingness characteristics, such as measurement availability, values within a specific range, or patient cohort.

References to Scenario 9 are presented in Section A.9.

Another common reason for data omission during pre-processing is the presence of invalid, unextractable, or erroneous values as stated by Scenario 10.

**Scenario 10** (Missingness of invalid data entries)**:** Data rows with invalid or erroneous entries are removed from the data during data pre-processing.

Examples are omission due to poor handwriting or corrupted medical chart pages [14], negative age values, or entries specified by ERROR code. References to Scenario 10 are presented in Section A.10.

### 3.3. Analysis of Missingness Scenarios

This section presents the foundation of the missing data theory necessary for analyzing the introduced scenarios in Section 3.2. After briefly explaining the missing data problem formulation, we describe the steps required for solving the problem under missing data, according to Figure 1. Throughout the steps presentation, we identify theoretical questions to answer for each scenario to bridge the gap between the analysis and application domains. We call these questions *inquiries* about the scenarios. Extensive details for the inquiries are presented in Appendix C.

#### 3.3.1. Setting and Notation

Let the random vector $X\in R^{d}$ comprise *d* study variables $X_{i}\in X,\;i\in \left\{1,\cdots ,d\right\}$. For ease of reference, we denote a specific variable of interest beside *X* (e.g., the class labels in the ML classification problem) as *Y*. Furthermore, we denote independence between $X_{i},X_{j}$ as $X_{i}⊥\;\;\;⊥X_{j}$. Independence by conditioning on a variable $X_{h}$ is denoted as $X_{i}⊥\;\;\;⊥X_{j}|X_{h}$.

In reality, the variable $X_{i}$ is realized for all patients, though it may or may not always be available (i.e., observed, recorded, and present in the dataset). We therefore refer to $X_{i}$ as a *counterfactual* variable since this is what the data would have been if they had always been available, possibly contrary to reality. Corresponding to each $X_{i}$, we define a binary variable $R_{i}\in \left\{0,1\right\}$, called the *missingness indicator*, to express $X_{i}$’s availability: we set $R_{i}=1$ when $X_{i}$ is available, and $R_{i}=0$ otherwise. The version of $X_{i}$ which is masked by missingness is called *proxy variable*, denoted as $X_{i}^{*}\in R\cup \left\{\mathrm{NaN}\right\}$ where NaN represents the missing entries. By this definition, a proxy variable is modeled as
(1)$$X_{i}^{*}=\left\{\begin{array}{cc} X_{i}, & R_{i}=1,\\ \mathrm{NaN}, & R_{i}=0. \end{array}\right.$$

The distribution of *R* is determined by the subset of scenarios from Section 3.2, which describe the data observation and recording in a healthcare facility and dataset selection for analysis. A data availability policy π represents the union of scenarios such that the missingness distribution follows the policy, i.e., R∼π. Subsequently, the resulting distribution given a policy π is denoted as $p_{\pi }$. We denote three special policies to reference in the paper: (1) the initial policy during data collection as $\pi_{\mathrm{init}}$, (2) the full-availability policy as $\pi_{\mathrm{full}}$, under which *X* is always available, and (3) any other policy as $\pi_{\mathrm{new}}$, which is neither $\pi_{\mathrm{init}}$ nor $\pi_{\mathrm{full}}$. This notation yields $p\left(X\right)=p_{\pi_{\mathrm{full}}}\left(X^{*}|R=1\right)$.

**Example 1** (missingness under availability policies)**.***Suppose a variable* $X_{1}$ *is realized for four patients, giving* $X_{1}={\left(2,3,4,7\right)}^{⊤}$. *If the fourth patient has missing values under a policy* $\pi_{\mathrm{init}}$, *we have* $X_{1}^{*}=(2,3,4,$ *NaN*)^⊤^. *In this case, the mean estimations for X and* $X^{*}$ *are defined as* $E_{\pi_{\mathrm{init}}}\left(X_{1}^{*}|R=1\right)=3$ *and* $E\left(X\right)=E_{\pi_{\mathrm{full}}}\left(X_{1}^{*}|R=1\right)=4$. *Under a new policy* $\pi_{\mathrm{new}}$ *where only the* X≥4 *observations are available, we have*  $E_{\pi_{\mathrm{new}}}\left(X_{1}^{*}|R=1\right)=5.5$

An availability policy for $X_{i}$ is, in general, parameterized by all variables (including $X_{i}$ itself) as well as other missingness indicators, i.e., $\left\{X,R_{\setminus i}\right\}$. To encode these dependencies, we model the joint (X,R) distribution using m-graphs Mohan et al. [2]. An m-graph under the availability policy π, denoted as $G_{\pi }\left(V\right)$, is a causal directed acyclic graph (DAG) with the node set $V=\{X\cup X^{*}\cup R\}$. The edges in the structure $X_{i}\to X_{i}^{*}\leftarrow R_{i}$ are deterministic, representing Equation (Equation 1). While non-graphical approaches for missing data exist, we focus on m-graphs for their effectiveness and popularity in this paper. Section 3.3.3 will provide more details about m-graphs.

Example illustrations of m-graphs are depicted in Figure 3, where three m-graphs model different policies for a similar $\left(X_{1},X_{2}\right)$ distribution.

#### 3.3.2. Defining the Estimand

In the first step of data analysis, an objective must be set by the domain expert and the data scientist and translated into an estimand, which will be fitted to the data. Examples include finding the weights of a prediction model for patient morbidity or the mean value of a biomarker for a population. Based on the form of the estimand, and whether and how it depends on the unavailable data distribution under missingness, we may face diverse challenges.

A basic question of interest is the mean of an outcome variable *Y* (e.g., the mean value of a test or the chance of recovery). If *Y* is partially available under the policy $\pi_{\mathrm{init}}$, one may formulate the question directly as $E_{\pi_{\mathrm{init}}}\left(Y^{*}|R_{Y}=1\right)$, which reads as the “mean of *Y* when it is available”. However, we are often interested in estimating the entire population regardless of the missingness status “had *Y* for all samples been available for analysis”. This objective, referred to as the counterfactual mean estimation, is presented as
(2)$$E\left(Y\right)=E_{\pi_{\mathrm{full}}}\left(Y^{*}|R_{Y}=1\right).$$

**Example 2** (counterfactual mean LDL cholesterol level)**.***As part of public health research*, *we aim to estimate the nationwide average LDL cholesterol level*, *denoted as Y*. *Available datasets are collected from a hospital where LDL levels are not available for all the patients*.

$E_{\pi_{\mathrm{init}}}\left(Y^{*}|R_{y}=1\right)$ *gives the average observed value in the hospital*.*As a possible new policy*, $E_{\pi_{\mathrm{new}}}\left(Y^{*}|R_{y}=1\right)$ *gives the average value if the LDL level had been observed for all patients in the hospital*.$E\left(Y\right)=E_{\pi_{\mathrm{full}}}\left(Y^{*}|R_{y}=1\right)$ *gives the target estimand, i.e., the nationwide average LDL level*.

As a more advanced objective, we may be interested in developing a prediction model for the outcome variable *Y* using the covariate vector *X*, i.e., E(Y|X=x), which reads as “conditional mean of *Y* given *X*”. We often choose an ML model for estimation, such as a multi-layer perceptron neural network f(x;w), parameterized by *w*. The weights of the network are learned by minimizing a loss function, e.g., the mean squared error (MSE): $E[{\left(y-f(x;w)\right)}^{2}]$. Model performance at deployment can also be evaluated using the same formula.

Given a fully observed outcome and missing covariates, the estimand
(3)$$E_{\pi_{\mathrm{init}}}\left[{\left(y-f(x^{*},r_{X};w)\right)}^{2}\right]$$
formulates the MSE loss for the available *X*. The estimand in Equation (Equation 3) suits the situation where the prediction model is to be deployed in an environment with the same observation policy, meaning that all missingness scenarios are the same during deployment as during the data collection stage. In Equation (Equation 3), we may use the information in $R_{X}$, e.g., we train (maximum) $2^{d}$ separate sub-models $g\left(x_{j}\right),R_{j}=1$ for each unique value (pattern) of *R* [20].

**Example 3** (Health status estimation at hospital discharge)**.***We aim to develop a prediction model for the 6-month outcome based on the observed variables during hospitalization*, *queried at discharge. The model deployment will not influence the physicians’ decisions*. *The fact that the hospitalization data are being analyzed retrospectively can justify the assumption that the observation and recording policy will not change at deployment*. *The MSE loss for this case is given by the estimand in Equation (Equation 3)*.

Alternatively, we may be interested in learning a prediction model that is deployed in healthcare facilities with different missingness scenarios, e.g., with varying guidelines of observation and protocols (Scenarios 5 and 6), for a different patient cohort (Scenarios 1 and 9), or in the same healthcare facility but with a change of observation policy because the physicians would measure different variables to “feed” it to the prediction model. In particular, suppose a training dataset generated given the m-graph in Figure 3a will be deployed in an environment modeled by the m-graph in Figure 3b. The estimand for such a case is
(4)$$E_{\pi_{\mathrm{new}}}\left[{\left(y-f(x^{*},r_{X};w)\right)}^{2}\right],$$
which reads as “MSE loss under new missingness scenarios at deployment”, where $\pi_{\mathrm{new}}$ represents the new policy.

**Example 4** (Change in hospital discharge protocols)**.***Suppose the hospital in Example Equation 3 adopts a new discharge protocol mandating performing a medical test for all patients before discharge*. *The MSE loss under the newly adopted policy is given by the estimand in Equation (Equation 4)*.

A special case of Equation (Equation 4) is when the prediction model is expected to make predictions always using full covariates (Figure 3c). The estimand for this case is $E_{\pi_{\mathrm{full}}}\left[{\left(y-f(x;w)\right)}^{2}\right]$, with only one missingness pattern, the full-availability $R=\vec{1}$. This objective is employed for most clinical prediction models (see Tsvetanova et al. [8]). For more examples, Appendix B presents the estimands for prediction using decision trees and feature importance.

**Example 5** (Clinical prediction model)**.***Suppose a clinical prediction model is developed using an incomplete dataset*. *As a result of successful development, physicians will use the model while they actively collect all study variables every time to feed to the model*. *The MSE loss at deployment is given by* $E_{\pi_{\mathrm{full}}}\left[{\left(y-f(x;w)\right)}^{2}\right]$.

#### 3.3.3. Identification

As shown in the previous step, estimands may query different missingness distributions, while the only available distribution is given by the data collection policy $\pi_{\mathrm{init}}$. If an estimand queries $\pi_{\mathrm{init}}$, such as Equation (Equation 3), it can be computed directly using the training dataset D. On the other hand, estimands such as (Equation 2) and (Equation 4) query different distributions and hence are subjected to the distribution shift problem. In the identification step, we find a procedure that computes a consistent estimate of an estimand under a target distribution using the available $\pi_{\mathrm{init}}$ [2].

To elaborate further, we consider an estimation approach under distribution shift, namely, *importance sampling*: for a functional θ of the distribution at deployment q(X,R), θ is estimated using the data collection distribution p(X,R) as
(5)∫θ(x,r)·q(x,r)dxdr=∫θ(x,r)·λ(x,r)p(x,r)dxdr,
where the fraction λ(X,R)=p(X,R)/q(X,R) is called the importance ratio. By Equation (Equation 5), samples are drawn from p(.) but re-weighted by their “importance” in reflecting q(.). Equation (Equation 5) states that a θ estimation is possible given the p(X,R) samples when λ is known for all (x,r) over the support of *p*.

The importance ratio can be re-written using the selection model factorization [42] as 
(6)$$\lambda \left(X,R\right)=\frac{q(X)q(R|X)}{p(X)p(R|X)}.$$

The conditional terms p(R|X) and q(R|X) in the fraction are the data collection and deployment availability policies, respectively. Assuming no additional counterfactual data distribution shift, i.e., p(X)=q(X), Equation (Equation 6) is simplified as λ(X,R)=q(R|X)/p(R|X), i.e., the ratio of missingness models at the data collection and deployment stages. When the availability policy does not change at deployment, the ratio is re-written as
(7a)$$\lambda \left(X,R\right)=\frac{p_{\pi_{\mathrm{init}}}\left(R|X\right)}{p_{\pi_{\mathrm{init}}}\left(R|X\right)}=1,$$
and when a new policy is adopted at deployment, it is re-written as
(7b)$$\lambda \left(X,R\right)=\frac{p_{\pi_{\mathrm{new}}}\left(R|X\right)}{p_{\pi_{\mathrm{init}}}\left(R|X\right)}.$$

While the following arguments are valid for Equation (Equation 5) in general, we consider a special case where the full-availability policy $\pi_{\mathrm{full}}$ is running at deployment (e.g., the estimand in Equation (Equation 2)). In this case, we trivially have λ(X,R)=0 for all incomplete data since the numerator $p_{\pi_{\mathrm{full}}}\left(X,R\right)$ is zero when $R\neq \vec{1}$. This means that only the complete cases (R = 1) are used for computation, for which $\lambda =1/p_{\pi_{\mathrm{init}}}\left(R=1|x\right)$. The resulting estimator according to Equation (Equation 5) is expressed (for estimation using D) as
(8)$${\hat{\theta }}_{\mathrm{IPW}}=\frac{1}{N}\sum_{X,R∼D}\theta \left(x,r\right)\cdot \frac{1\left(r=1\right)}{p_{\pi_{\mathrm{init}}}\left(R=1|x\right)},$$
for *N* samples, where 1(r=1) selects only the complete cases. Equation (Equation 8) is known as the *inverse-probability weighting* (IPW) estimator. The denominator in Equation (Equation 8) is referred to as the propensity score, often denoted as PS(x).

The challenge of identification lies in the conditioning set of importance ratio terms, as they generally depend on the counterfactual distribution (X), which is only partially available. As a solution, we assume an m-graph for the problem and seek independence properties among (X,R) variables that allow us to express the importance ratio in terms of factors that can be estimated using the available distribution $\left(X^{*},R\right)$. For the scope of this paper, we mainly focus on identification with respect to m-graphs [2,43]. See Section 3 of Mohan et al. [2] for other identification approaches.

**Example 6** (Identification with respect to an m-graph)**.***Suppose a functional* $\theta (X_{1},X_{2})$ *is to be estimated, given the data collection and deployment policies* $\pi_{\mathrm{init}}$ *and* $\pi_{\mathrm{full}}$, *respectively. The propensity score for IPW estimator is* $p_{\pi_{\mathrm{init}}}\left(R_{1}=1,R_{2}=1|X_{1},X_{2}\right)$, *which cannot be estimated using*  D. *Assuming the m-graph in Figure 3a, we proceed as follows (we drop the distribution index for brevity)*:


*Factorize*: $\mathrm{PS}\left(X_{1},X_{2}\right)=p\left(R_{1}=1|X_{1},X_{2}\right)p\left(R_{2}=1|R_{1}=1,X_{1},X_{2}\right)$*The assumed m-graph gives* $R_{1}⊥\;\;\;⊥X_{1},X_{2}$ *and* $R_{2}⊥\;\;\;⊥X_{2}|R_{1},X_{1}$. *The propensity score is thus rewritten as* $p\left(R_{1}=1\right)p\left(R_{2}=1|R_{1}=1,X_{1}\right)$*By the missingness definition in Equation (Equation 1), we express the second term using the proxy variable and rewrite the propensity score as* $p\left(R_{1}=1\right)p\left(R_{2}=1|R_{1}=1,X_{1}^{*}\right)$.


*Both factors in the propensity score can be estimated using* $\left(X^{*},R\right)$

In conclusion, identification in this manner requires an m-graph model, and within it, the causal relations of the missingness indicators are specifically important. It is therefore necessary to discover what kind of causal structures the missingness scenarios induce for *R*. In particular, we specify the parents and ancestors (direct and indirect causes) for the *R* nodes as stated by Inquiry 1. The causes of *R* nodes are commonly referred to as the missingness mechanism.

**Inquiry 1** (missingness mechanism)**:** Causal relations that a scenario implies for *R* nodes.

To facilitate identifying the causes, we categorize all potential causes to search for in the following three categories:
(*X* and *R* components) First, the candidates for causes of *R* are the study variables and their corresponding missingness indicators within the dataset. Examples can be found in Figure 3a,b, where *X* is a cause of indicator *R*.(latent/hidden confounders) Variables that have not been collected and available in the dataset may also causally influence *R*. More importantly, they may confound two or more study variables within the estimand, and may therefore hinder the identification process.(exogenous causes) Other variables that may lie outside the dataset and do not confound the study variables of interest are considered exogenous causes, having, in general, no identification implications.

Missingness in health-related variables such as lab test items is mainly caused by physicians under Scenario 5 (missing due to diagnostic irrelevance), where they make measurement decisions based on the observed history. Therefore, in this case, *R* indicators for health variables have incoming edges from the previous measurements (recorded or unrecorded). Other potential causes include the health status under Scenarios 1 (patient complete non-visit) and 2 (missing follow-up visit due to health status). Examples of the latent/hidden confounders include socioeconomic variables as well as variables in secondary datasets with information about the non-visit population under Scenario 1. As for the exogenous causes, many causes may be recognized, such as simply forgetting to enter the data for a patient, under Scenario 8 (unrecorded observations). However, one should be cautious about treating all medically unrelated variables as exogenous causes, as they may still confound the study variables and missingness indicators. A detailed analysis of missingness scenarios with respect to Inquiry 1 is presented in Table A1.

Inquiry 1 explores the structural distribution shift caused by a change in the m-graph between the data collection and deployment stages. Another possibility is that the m-graph stays invariant, but the causal relations are subjected to a parametric shift. For example, assume the m-graph in Figure 3a holds for both data collection and deployment, but the missingness probability in $X_{2}$ changes from $\sigma (R_{1}+2X_{1})$ to $\sigma (0.2R_{1}+5X_{1})$. As stated by Inquiry 2, it is crucial to explore the potential parametric shift at deployment due to a change in the observation and recording policies.

**Inquiry 2** (Missingness distribution shift)**:** Whether a scenario is subjected to missingness parametric distribution shift at deployment.

Parametric shift may occur in Scenario 5 (missing due to diagnostic irrelevance), if the definition of normal/abnormal ranges for a health marker changes. In this case, the results of primary tests still influence the performing decision of later tests, however, via different rules. As another example, a parametric shift may occur in Scenario 7 (missing due to resource unavailability), if the monetary cost of a medical test decreases as a result of equipment upgrade or insurance plans, leading physicians to order the test more often. A detailed analysis of missingness scenarios with respect to Inquiry 2 is presented in Table A2.

**Example 7** (parametric shift due to decreased test costs)**.***Consider a primary test* $X_{1}$ *and a secondary and more expensive test* $X_{2}$. *Patients with abnormal primary test values* ($X_{1}>5$) *are more likely to give the* $X_{2}$ *test*. *After a cost reduction for the* $X_{2}$ *test*, *the overall frequency of the test* ($R_{2}=1$) *increases such that now the relative number of tests for patients with abnormal* $X_{1}$ *values is* ρ *times larger than before, yet the association between* $X_{1}$ *and* $R_{2}$ *is retained*. *This statistics gives*$$\frac{p_{\pi_{\mathrm{new}}}\left(R_{2}=1|X_{1}>5\right)}{p_{\pi_{\mathrm{init}}}\left(R_{2}=1|X_{1}>5\right)}=\rho ,$$*which is the importance ratio for* $\left(X_{1}=1,R_{2}=1\right)$ *samples in Equation (Equation 5)*. *We leave it to the readers to calculate other importance ratios based on assumed statistics about this hypothetical problem*.

So far, the described identification methodology has been based on the selection model factorization in Equation (Equation 6) and the no-distribution-shift assumption for the counterfactual variables. However, there might exist missingness scenarios under which this assumption is violated. A case of violation is when the observation and measurement decisions directly affect the counterfactual variables. In terms of m-graphs, this translates to an R→X edge. The assumption that such a causal relation does not exist is referred to as no-direct-effect (NDE) [22], discussed in the m-graph identifiability literature [25]. Since the violation of NDE influences the identification procedure, it is crucial to know whether the problem setting permits it as stated by Inquiry 3.

**Inquiry 3** (no-direct-effect assumption)**:** Whether a scenario implies outgoing edges from missingness indicators to counterfactual variables.

A crucial case of NDE violation occurs when invasive tests such as biopsy affect the health status of patients. The effect of observation may be exerted on the corresponding counterfactual variable itself or other variables. This effect may also be exerted indirectly, e.g., through temporarily stopping a certain medication before a medical test. For example, due to the contraindication of radiology contrast agents and metformin, it is recommended that for diabetic patients, medication is stopped before performing angiography [44]. Note that under violation of the NDE assumption, the problem definition stated in Section 3.3.2 becomes ill posed and requires further elaborations. An example of a problem definition under NDE violation is discussed in Example 8. A detailed analysis of missingness scenarios with respect to Inquiry 3 is presented in Table A3.

**Example 8** (Problem definition under NDE violation)**.***Assume the following m-graph* $X\to Y\leftarrow R_{X}$, *describing an* (X,Y) *dataset with fully observed Y*, *where the measurement of X negatively influences Y*. *This problem cannot be analyzed unlike Example 1*, *as the counterfactual realizations cannot be described ignoring the missingness status. As a hypothetical data generation mechanism, suppose the *X−Y *relation follows* Y=wX+ϵ,ϵ∼N(0,1) *in the absence of any measurement* ($R_{X}=0$). *When* *X* *is measured* ($R_{X}=1$), *the Y distribution changes to* $Y=wX+w_{0}+\epsilon$. *Therefore*, $E_{\pi_{\mathrm{init}}}\left(Y\right)\neq E_{\pi_{\mathrm{full}}}\left(Y\right)$. *Possible questions to pose with regard to a target quantity* θ *are as follows*:



*If the observation policies remain unchanged;*
*If we begin to always observe X*;*If we knew the value of X but without negative influences on Y*, *e.g., using a new testing technology*.


Another common assumption for the missing data problem is the no-interference assumption, stating that the measurement decisions for one individual do not affect other individuals [22]. This is similar to the independent and identically distributed assumption in general ML problems: having interfered measurements, the independent and identically distributed assumption cannot be made for the *R* distribution. It is therefore important to check whether the no-interference assumption is permitted for observation scenarios as stated by Inquiry 4.

**Inquiry 4** (No-interference assumption)**:** Whether a scenario causes interference among the availability status of data samples.

Similar to the NDE assumption, one may find realistic scenarios where the no-interference assumption is violated. In general, competing for limited resources under Scenario 7 (unavailability or shortage of resources) or for available hospitalization services under Scenario 1 and 2 (complete non-visit and missing follow-up) imply interference. A detailed analysis of missingness scenarios with respect to Inquiry 4 is presented in Table A4.

Finally, we discuss a unique case of missingness, where data samples are completely omitted from the dataset prior to any analysis. This case can be modeled in m-graphs via an $R^{†}$ node that influences all $R_{i}$ such that if $R^{†}=0$, then $R_{i}=0,\forall i$ (Figure 4). The risk in this situation lies in the fact that we cannot infer the occurrence of such omissions from a dataset without additional information, which may thus lead to the wrong conclusion that the dataset is complete and free of missingness. This case is commonly referred to as *selection bias* in causal inference literature. Selection bias is argued in Inquiry 5.

**Inquiry 5** (Selection bias)**:** Whether a scenario causes the omission of an entire data sample in the form of selection bias.

Clearly, sample omission can be a result of non-visit under Scenario 1 and inclusion/exclusion criteria under Scenario 9. Whether or not this should be conceived as a bias depends on whether the target parameter (e.g. *Y* in Equation Equation 2) is believed to vary between the observed and the unobserved sub-populations. A detailed analysis of missingness scenarios with respect to Inquiry 5 is presented in Table A5.

#### 3.3.4. Estimation

There are several methods for estimation with missing data, including likelihood-based methods such as the Expectation Maximization (EM) algorithm, multiple imputation (MI), IPW estimator, and outcome regression (OR) [27,42]. In the scope of this paper, we continue with the importance sampling approach in Equation (Equation 5), in particular, the IPW estimator in Equation (Equation 8) and the estimation of the propensity score.

Even though a successful identification step guarantees that the propensity score can be estimated using the available data, we still face some challenges, e.g., when the missingness pattern is *non-monotone*. A missingness pattern is called monotone if there is at least one ordering of the variables such that observing the *j*-th variable ensures that all variables k>j in the ordering are all observed for all samples (Figure 5a). Estimation of the propensity score has a straightforward solution for monotone patterns. Example 9 showcases propensity score estimation for identifiable monotone missingness.

**Example 9** (Propensity score estimation for identifiable monotone missingness)**.***For the missingness in Figure 5a, we have* $\sum_{j=1}^{4}p\left(S_{j}|X\right)=1$, *while* $\mathrm{PS}\left(X\right)\equiv p\left(S_{1}|X\right)$. *Assuming identifiability, each* $p(S_{j}|X)$ *can be estimated using only the variables available in* $S_{j}$. *As a result, the propensity score is estimated as*(9)$$\mathrm{PS}\left(X\right)=1-p\left(S_{2}|X_{1},X_{2},X_{3}\right)-p\left(S_{3}|X_{1},X_{2}\right)-p\left(S_{4}|X_{1}\right).$$

While methods have been developed for an effective estimation under non-monotone missingness [27,31], it is beneficial to adopt monotone solutions if applicable. In that regard, Inquiry 6 argues whether a missingness scenario individually induces monotone missingness patterns.

**Inquiry 6** (Monotonicity)**:** Whether a scenario induces missingness with monotone patterns.

If an individual missingness scenario is active, monotonicity can be directly inferred from the emerged patterns, revealed by a simple sorting of the variables with respect to their missingness ratio (Figure 5a). However, in practice, several scenarios influence a dataset. In such cases, the monotone pattern attributed to one scenario is broken by other scenarios. If we can attribute the emerged non-monotone pattern to a dominant monotone-inducing scenario along with less effective non-monotone scenarios (hypothetically in Figure 5b), then methods exist based on resolving the missing entries up to recovery of the monotone pattern, e.g., via imputation, and proceeding with IPW estimation for monotone missingness [45]. A noteworthy scenario likely inducing monotonicity is the sequential observations of physicians under Scenario 5 (missing due to diagnostic irrelevance). Given a specific diagnostic flowchart, it is reasonable to assume that more specific secondary tests shall not be made unless primary tests are conducted. As said, this pattern may be broken for many reasons, including more than one diagnostic flowchart being used and other scenarios such as 4 (patient’s refusal) or 7 (resource unavailability). A detailed analysis of missingness scenarios concerning Inquiry 6 is presented in Table A6.

#### 3.3.5. Sensitivity Analysis

The assumptions made for handling missing data may not hold under all circumstances. They might be too strong for practical implementation, or we may expect the environment to undergo some perturbations that violate them. To ensure the robustness of the analysis, it is crucial to measure the sensitivity of results to departures from the assumptions and report the variation. Sensitivity analysis is usually performed by perturbing the m-graph model.

In addition, it is possible that due to the nature of the problem, assumptions do not lead to a successful identification. In this case, we may impose stronger assumptions that lead to identifiability, model the departures from the actual assumptions, and finally measure the sensitivity to different degrees of magnitude of those departures.

**Example 10** (Sensitivity analysis for the unidentifiable self-masking missingness)**.***Consider an outcome variable Y that is subjected to missingness under the following mechanism* $Y\to R_{Y}$. *The estimand* E(Y) *is unidentifiable under this mechanism, referred to as self-censoring [25] or self-masking [20].*

*We can assume that the mean of the unobserved population is* δ *units away from the observed population, additively* $E_{\pi_{\mathrm{init}}}\left(Y|R=0\right)=E_{\pi_{\mathrm{init}}}\left(Y^{*}|R=1\right)+\delta$, *or multiplicatively* $E_{\pi_{\mathrm{init}}}\left(Y|R=0\right)=\delta E_{\pi_{\mathrm{init}}}\left(Y^{*}|R=1\right)$ [6,33]. *We then measure the variation of* $E_{\pi_{\mathrm{full}}}\left(Y^{*}|R=1\right)$ *assuming a range of values for the sensitivity parameter* δ.

For reliable meaningful sensitivity analysis results, it is crucial to interpret the sensitivity parameters based on meaningful real-world quantities. Inquiry 7 states that scenarios may carry valuable information for choosing meaningful parameters.

**Inquiry 7** (Meaningful sensitivity parameters)**:** Given a scenario, what are the meaningful units and ranges of parameters for sensitivity analysis.

Specific to the importance sampling approach and Equation (Equation 8), the unidentifiable terms appear in the importance ratio. The importance ratio captures the differences in the levels of availability for different covariate strata. To make an informed guess about this quantity, we may refer to other research works or collaborations with health domain experts. For instance, Zamanian et al. [36] suggest that the sensitivity parameters for physicians’ observations (Scenario 5) are related to the odds of making an observation for relatively healthy or sick patients, which can be inferred based on the guidelines, protocols, and referring to the attending physicians. The sensitivity parameter for this case is formulated for the model in [36], assuming a logistic model for missingness, by the following odds-ratio term:(10)$$\delta =\log\frac{O(R|\mathrm{healthy})}{O(R|\mathrm{sick})},$$
where O(R|X)=p(R=0|X)/p(R=1|X). Equation (Equation 10) follows the so-called exponential tilting model, where the multiplicative departure for *R* is modeled as an exponential term [34].

Likewise, the parameters for hospital visits (Scenario 1) are related to the odds of visiting a healthcare facility for healthy and sick populations, formulated similarly as Equation (Equation 10), for which some information can be extracted from epidemiologic studies and public health reports. Overall, the form of the sensitivity model depends on the estimand and the estimator. Yet, a similar ratio as in Equation (Equation 10) often appears in the analysis, which must be specified. A detailed analysis of missingness scenarios concerning Inquiry 7 is presented in Table A7.

## 4. Experiment

### 4.1. Setup

In this section, we perform a simulation experiment to demonstrate the impact of different scenarios and their corresponding assumptions on the results of the missing data analysis. Simulation allows us to induce different missingness scenarios in a controlled manner and evaluate the reliability of analysis compared to the ground truth. Details regarding the study design and implementation of simulations, medical use case, missingness scenarios, estimands, algorithm derivations, and missing data methods are presented in Appendix D.

Figure 6 presents three m-graphs with a shared counterfactual causal DAG for cardiovascular disease (CVD) influenced by baseline variables and measurements, namely age, body mass index (BMI) at admission, (systolic) blood pressure (BP), and CVD outcome. The causal DAG was inspired by the work of Bakhtiyari et al. [46].

For the CVD causal model, three combinations of missingness scenarios were induced:Case 1:(Figure 6a) We induced Scenario 10, where invalid entries were dropped from the dataset.Case 2:(Figure 6b) In addition to Scenario 10, we induced Scenario 1, where the healthy sub-population in terms of CVD visited the healthcare facility less frequently.Case 3:(Figure 6c) In addition to Scenario 10, we induced Scenario 5, assuming that age and BMI values were always measured and influenced the measurement of BP while they may have not been always recorded (Scenario 8).

For each case, we estimated two objective estimands:Estimand 1:(Counterfactual mean BP) We estimated the counterfactual mean blood pressure.Estimand 2:(Classification accuracy for CVD) We evaluated the MSE loss for a trained prediction model for CVD under the full-availability policy. The estimand was $E_{\pi_{\mathrm{full}}}\left[{\left(y-f(x)\right)}^{2}\right]$. The model f(.) was a logistic regression classifier trained on a mean-imputed dataset. The focus of this study was only to estimate the performance of the existing model at deployment. Therefore, perfect model training or fine-tuning was not necessary.

Finally, the analysis was performed using three different missing data methods:Method 1:We performed complete-case analysis (CCA). For the mean BP estimand, we took the average BP value for the observed cases. For the model performance estimand, we tested f(.) using complete cases only.Method 2:We used the Missforest imputation method [29] for the incomplete datasets. The analyses were then carried out using the imputed dataset.Method 3:We performed IPW estimation using Equation (Equation 8). After the identification step, propensity scores were estimated via re-weighted complete cases.

We modeled the structure of the m-graphs for each case and simulated their distribution parameters in 20 iterations by sampling the uniform randomly from a search space. In this way, various simulation cases were generated, following the imposed structure [47]. For each iteration, we reported the estimation bias regarding absolute error using the ground truths, namely, the counterfactual mean BP (on a log scale) and factual classification accuracy.

### 4.2. Results

Figure 7a presents the estimation bias for the estimands 1 and 2 on a log scale. For method 1 (CCA), only case 1 aligns with its assumption, giving the most negligible bias among the three cases. For method 2 (Missforest), cases 1 and 3 align with their assumption, yet interestingly, case 3 has the highest bias. Method 3 (IPW) is designed to be domain-informed. Higher biases in case 3 for methods 2 and 3 can be attributed to possible model misspecifications for the propensity score estimation. Overall, the domain-informed IPW estimation gives the lowest bias.

Figure 7b presents the results for the classification accuracy estimation at deployment. The average classification accuracy of the logistic regression model was 0.77 across all cases. Throughout the experiment, three methods suffered from approximately up to 0.1 bias (13% of the average accuracy), except for the CCA estimation in case 2 with up to 0.25 bias (33% of the average accuracy).

The experiments reflect an important message of this paper, that estimation bias may considerably vary for missing data methods under different circumstances, while it cannot be discovered during the training phase.Therefore, the only trustworthy approach to missing data is to rely on realistic assumptions and domain-informed methods.

## 5. Discussion

In order to overcome the challenges of a missing data problem, it is recommended to report the problem transparently [3,4], collect complementary datasets from various sources [7,14], and choose a suitable method accordingly. However, the question remains: what kind of data should we collect and incorporate into the analysis, and how should the missing data be reported?

This paper propounds a framework for analyzing missingness in observational health data. Via the recognition and categorization of fundamental scenarios, we provide a basis for understanding the data collection processes in healthcare facilities. According to this framework, focusing on human agents (physicians, patients) and environments (hospitals, clinics) helps discover and report the scenarios. As shown in this paper, the implications of scenarios must be formulated for identification, estimation, and sensitivity analysis steps. Once the corresponding assumptions are specified, a suitable method can be selected. During the analysis, complementary datasets can be utilized when they enable and facilitate the analysis steps, e.g., when new variables make the missingness mechanism identifiable (Inquiry 1) or when a dataset offers meaningful interpretations of the sensitivity parameters (Inquiry 7).

This paper is intended for not only the medical data scientists but also the developers of missing data methods in the ML community as a response to the recurring theme of devising sophisticated (deep learning) imputation models [48,49,50]. While imputation (or IPW) methods with high learning capabilities may considerably enhance the estimation step, most assumptions and inquiries concern the identification and sensitivity analysis steps (Inquires 1–5 and 7). Therefore, to ensure the method’s effectiveness, it is necessary to actively scrutinize the assumptions, especially if the application domain is safety-critical.

In fact, methodology papers often report the conditions using Rubin’s at-randomness categorization [23], according to which a missingness mechanism can be completely-at-random (MCAR), at-random (MAR), and not-at-random (MNAR). However, this categorization has been a blessing and a curse; even though it effectively formalizes the first step of identification addressed by Inquiry 1, it was introduced in the original work specifically for Estimand (Equation 2) and was later extended for the joint-distribution estimand p(X,Y) [25]. The works we criticize often make at-randomness assumptions disconnected from the reality of the use case and denuded of their context, ignoring the requirements of the estimand. If these works were more sensitive to the nuances of the problem and adhered to the standard missing data analysis process, they would positively influence the medical data analysis works that adopt them. We suggest that future works discuss the assumptions in a form similar to the inquiries in this paper and present some real-world examples for validity and violation cases.

### Limitations and Future Works

The current paper studied ten fundamental and prevalent scenarios considering healthcare facilities. Nevertheless, new and different scenarios may arise under new data observation, recording, and collection circumstances. Likewise, the inquiries in this paper were made regarding the general frame of the missing data theory, which is relevant for most analysis methods. It is conceivable that specific methods have unique assumptions and, therefore, further inquiries to explore. Nevertheless, the analysis process in this paper can be applied to new scenarios and inquiries.

The scope of this paper is limited to observational data from healthcare facilities. Other similar health domains, such as medical wearable sensors, can be similarly analyzed by employing the analysis framework presented in Section 3.1, e.g., one can study the scenarios induced by patients (e.g., missing measurement due to taking off the smartwatch because of irritation) or technical issues (operating system issues or low battery), and subject them to the same inquiries introduced in this paper.

In addition, exploring all the introduced scenarios scenarios in great detail would have indeed exceeded the publication’s scope. While the answers to the inquiries presented in Section 3.3 and Appendix C can be considered a constructive first step, future research endeavors will benefit from deeper investigations into each individual missingness scenario with greater granularity.

Furthermore, it is worthy of note that the missing data problem lies in the broader category of the data-coarsening processes, where the data veracity, representativeness, or completeness is impaired. Other data-coarsening processes in healthcare include the patient recall bias [51], variations in medical outcome definition over time or depending on the defining consortium [52], or coarsened representation of pertinent negative/positive values [14,53]. Extending our analysis framework to other data-coarsening problems in future works could bridge further translational gaps in medical and healthcare data analysis.

Finally, obtaining a real-world healthcare facility dataset with reliable ground truth for missing data is exceedingly challenging. Collecting such a dataset would require the investigation of all missingness occurrences without disrupting the environment and the missingness distributions. This is why we relied on the simulation experiment to be able to investigate different aspects of various objectives, scenarios, and methods. Nonetheless, we would obtain more insightful results if real-world datasets exist for validation, however small scale or simple.   

## Figures and Tables

**Figure 1 jpm-14-00514-f001:**
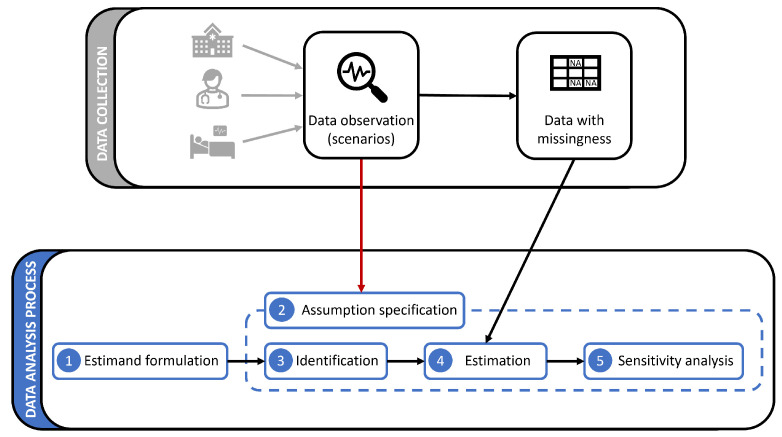
The ideal analysis process under missing data, consisting of (1) estimand formulation, (2) assumption specification, (3) identification, (4) estimation, and (5) sensitivity analysis.

**Figure 2 jpm-14-00514-f002:**
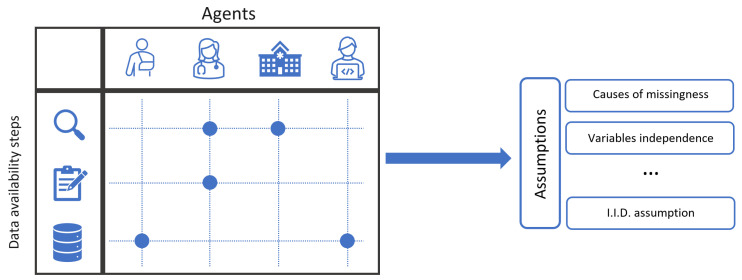
A schematic of the analysis framework in this paper: on the left, a cross-investigation of the data availability steps and agents guides us toward the existing scenarios (blue dots). Each scenario then will be subject to inquiries about various missing data assumptions on the right.

**Figure 3 jpm-14-00514-f003:**
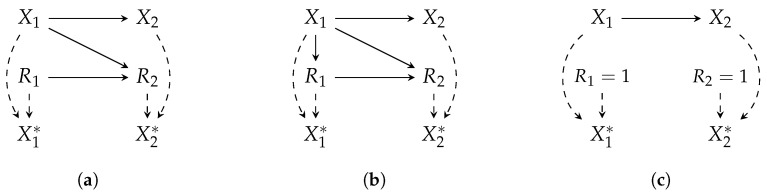
Three example m-graphs, modeling the joint distribution $\left(X,X^{*},R\right)$ for a bivariate dataset: (**a**) $\left(R_{1},R_{2}\right)∼\pi_{\mathrm{init}}\equiv \left(\pi_{\mathrm{init},1},\pi_{\mathrm{init},2}\left(R_{1},X_{1}\right)\right)$ represents the missingness distribution induced by missingness scenarios at data collection; (**b**) $\left(R_{1},R_{2}\right)∼\pi_{\mathrm{new}}\equiv \left(\pi_{\mathrm{new},1}\left(X_{1}\right),\pi_{\mathrm{new},2}\left(R_{1},X_{1}\right)\right)$ represents a new missingness distribution due to a change in the scenarios for $R_{1}$; (**c**) $\left(R_{1},R_{2}\right)∼\pi_{\mathrm{full}}\equiv \left(1,1\right)$ represents the full-availability case (no missingness scenario). Dashed edges are deterministic, encoding the definition in Equation (Equation 1).

**Figure 4 jpm-14-00514-f004:**
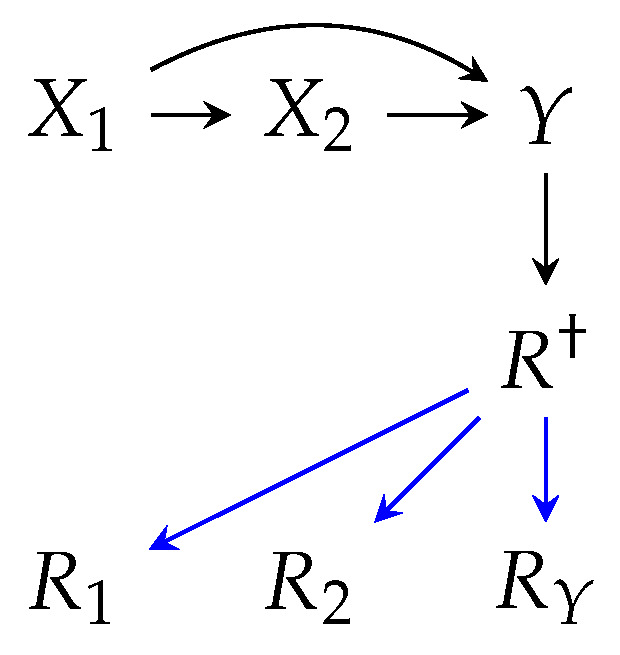
An example m-graph to model selection bias, determined by the *Y* values. Blue edges represent the deterministic masking relation between $R^{†}$ and $R_{i}$s.

**Figure 5 jpm-14-00514-f005:**
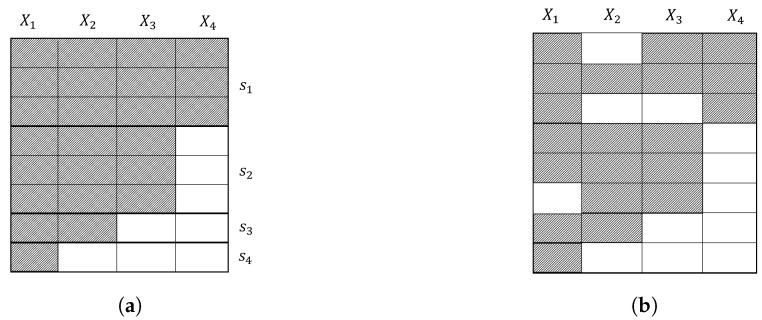
Two missingness patterns: (**a**) monotone; (**b**) non-monotone. The monotone pattern is described by four $S_{1}-S_{4}$ patterns. One can infer, using prior knowledge, that the non-monotone pattern in (**b**) is a result of some non-monotone missingness scenarios, interrupting the monotone pattern in (**a**).

**Figure 6 jpm-14-00514-f006:**
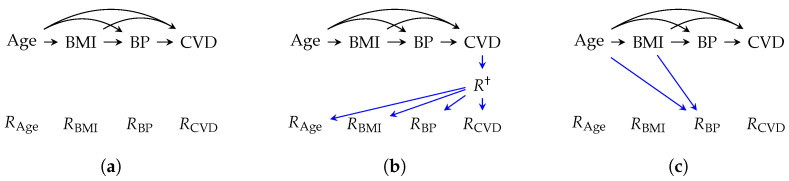
Example m-graphs for cardiovascular disease under the influence of different constellations of missingness scenarios: Scenario 1 (patient complete non-visit), 5 (missing measurements due to diagnostic irrelevance), 8 (unrecorded observations), and 10 (omission of invalid data entries). Given these scenarios, the three m-graphs induce the following: (**a**) Scenario 10; (**b**) Scenario 1 + 10; (**c**) Scenario 5 + 8 + 10. Top row nodes and edges are the shared counterfactual sub-graph. Blue edges encode the missingness mechanisms.

**Figure 7 jpm-14-00514-f007:**
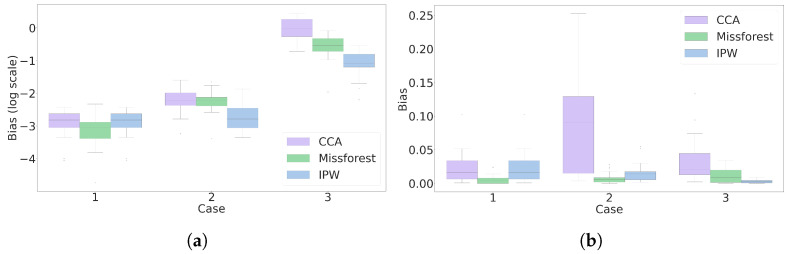
Estimation bias results for two objective estimands: (**a**) the counterfactual mean BP, (**b**) classification accuracy of the logistic regression model for CVD. Cases and methods are presented as the x-axis ticks and box plot colors, respectively.

**Table 1 jpm-14-00514-t001:** Overview of missingness scenarios in healthcare facilities, analyzed in this paper.

Scenario No.	Title
Scenarios related to patients	
1	Patient complete non-visit
2	Missing follow-up visit due to health status
3	Missing measurements due to health-related events during hospitalization
4	Missing measurements due to patient’s refusal
Scenarios related to physicians	
5	Missing measurements due to diagnostic irrelevance
Scenarios related to healthcare facilities	
6	Missing measurements outside protocols requirements
7	Unavailability or shortage of resources
8	Unrecorded observations
Scenarios related to data pre-processing	
9	Omission of data samples based on inclusion/exclusion criteria
10	Omission of invalid data entries

**Table 2 jpm-14-00514-t002:** Overview of theoretical inquiries for missingness scenarios analyzed in this paper.

Inquiry No.	Description
At identification step	
1	What missingness mechanism is induced by a scenario
2	Whether a scenario is subjected to missingness parametric distribution shift
3	Whether a scenario permits no-direct-effect assumption
4	Whether a scenario permits no-interference assumption
5	Whether a scenario induces selection bias
At estimation step	
6	Whether a scenario induces monotone missingness patterns
At sensitivity analysis step	
7	Whether a scenario gives informed guesses about sensitivity parameters

## Data Availability

The data used in the Experiment section were synthesized using the PyPARCS Python library for causal simulation. The simulation configuration is presented in Appendix D.

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
