# Peer review of "Analysis of Missingness Scenarios for Observational Health Data"

_jpm, 2024, doi:10.3390/jpm14050514_

Round 1

Reviewer 1 Report

Comments and Suggestions for Authors

The topic is exciting and important. However, several key areas need more work before publication. I have summarized the required changes in the hope that the feedback will be helpful to you as you update the paper.

1. The author should strive for greater transparency in reporting assumptions related to missing data in healthcare. Providing clear and detailed descriptions of the missingness problem statements and assumptions will enhance the credibility and reproducibility of the research.

2. While the paper identifies ten fundamental missingness scenarios in observational healthcare data, further exploration and analysis of each scenario with greater granularity could provide deeper insights and contribute to a more comprehensive understanding of missing data issues in healthcare.

3. The authors should consider expanding the analysis framework to include other health domains beyond observational data from healthcare facilities. Exploring missing data issues in areas such as medical wearable sensors and addressing scenarios induced by patients or technical issues can broaden the applicability of the research findings.

4. The authors should extend the analysis framework to encompass other data coarsening processes in healthcare. These processes, such as patient recall bias or variations in medical outcome definitions, can help bridge translational gaps in medical and healthcare data analysis. This expansion could lead to a more comprehensive understanding of data quality issues in healthcare research.

Reviewer 2 Report

Comments and Suggestions for Authors

The proposal presents an Analysis of Missingness Scenarios for Observational Health Data. The proposal presents an interesting topic; however, the following aspects were identified:

1.     The introduction indicates that the proposal presented is inspired by the work of Moreno-Betancur et al. [17] and Marino et al. [18]; however, it does not specifically indicate the main aspects that served as motivation for each work, nor does it indicate the main differences with respect to them.

2.     Likewise, it is important that the scientific contribution be clear and precise for both the health area and the computational area.

3.     It is suggested that in order to highlight the importance and give more relevance to the problem, worldwide statistics should be included and described, which allow visualizing the growth and importance of addressing the problem of loss of health data on which this proposal is based.

4.     In te section 4.            "Experiments" indicates the activities performed in the two experiments, however, the results obtained are not shown and described. That is to say, there is no reliable evidence of the realization of the proof of concept that validates the proposal presented. In addition, it is important and necessary to indicate, describe and discuss the results obtained in the experiments in the case of a possible replication of the experiments and even improvements that can be generated by other researchers.

5.     In addition, it would be relevant to indicate the bias of the simulations in the two experiments performed and it is suggested to include as future work to test and validate the proposal in real scenarios.

Comments on the Quality of English Language

NA

Reviewer 3 Report

Comments and Suggestions for Authors

The article submitted for review is quite interesting and could contribute to the discussion on the application of artificial intelligence in clinical practice. In order to improve the quality of the text, it is worth considering introducing several modifications:

Since the work is theoretical (analytical) in nature, it is worth emphasizing how the literature for the presented research was selected. Such a procedure is nothing more than a methodological requirement of scientific work, firstly, and secondly, it can help the reader determine the limitations of the conducted research. The explanations presented in lines 97-103 are not sufficient from a methodological point of view. For the clarity and coherence of the manuscript's content, it would be beneficial to add a section titled "Methods" or "Methodology" and describe it in detail.

It is not clear from which databases of scientific publications the authors drew. Therefore, it is difficult to identify and assess whether the selection of cited literature is appropriate and sufficient in each case. In the "Methods" section, it would be useful to indicate the criteria based on which the cited publications were included and excluded, as well as specify which scientific databases were used.

The limitations should be separated from the Discussion (See lines 694 and following...) and elaborated upon in more detail.

Reviewer 4 Report

Comments and Suggestions for Authors

Dear authors:

Thank you for submitting your article to JPM Journal. The manuscript highlights the gap for observational data from health-care facilities. The gap is identifying by ten fundamental missingness scenarios arising during data measurement, recording, and pre-processing in observational health data, influenced by physicians, patients, healthcare facilities, and data scientists.

The manuscript presents a very particular structure, considering that it is a “research article”. Perhaps, its understanding would require more visual support to complement the reviewed one. However, the article is considered appropriate and is well written.

Regardless of the technical or scientific sound, it is estimated that a better presentation and organization would greatly help the visualization and understanding of the content. Additionally, titles/subtitles/sections are not presented in a standardized form, which contribute to reading dispersion.

Please, improve the structure and writing of the abstract. Try to describe the content in a deep technical and cohesive way. Additionally, I kindly request for an orderly restructuring of the work (principally the introduction) and a clear definition of the limitations.

Zamanian, A. and Ahmidi, N., to avoid self-citation, do not include more than two of your references in the work. Mainly, consider published works.

Thank you in advance.

Round 2

Reviewer 2 Report

Comments and Suggestions for Authors

The authors correctly addressed the observations of the review.

Comments on the Quality of English Language

NA